# Reopening of Schools in the COVID-19 Pandemic: The Quality of Life of Teachers While Coping with This New Challenge in the North of Spain

**DOI:** 10.3390/ijerph18157791

**Published:** 2021-07-22

**Authors:** Nahia Idoiaga Mondragon, Naiara Berasategi Sancho, Maria Dosil Santamaria, Naiara Ozamiz-Etxebarria

**Affiliations:** 1Department of Evolutionary and Educational Psychology, University of the Basque Country UPV/EHU, 48940 Leioa, Spain; nahia.idoiaga@ehu.eus; 2Department Didactics and School Organisation, University of the Basque Country UPV/EHU, 48940 Leioa, Spain; naiara.berasategi@ehu.eus; 3Department of Research and Diagnostic Methods in Education, University of the Basque Country UPV/EHU, 48940 Leioa, Spain; maria.dosil@ehu.eus

**Keywords:** COVID-19, education, health, quality of life, teachers

## Abstract

Background: This study aims to analyze how teachers perceived their quality of life when coping with the reopening of schools after their closure due to the COVID-19 pandemic. Methods: This study was carried out with a total sample of 1633 teachers from the Department of Education of the Basque Autonomous Community (Northern Spain), all of the professionals working in different educational centers, from preschool education to university studies, with the average age of 42.02 years (SD = 10.40). Main Outcome Measures: For this purpose, the Spanish version of the WHOQOL-BREF was used. Results: The highest values of perceived quality of life were found in the dimension of psychological health, followed by the dimension of physical health, the social relations dimension, and finally, the environmental dimensions. The results also revealed significant differences depending on gender, age, having a chronic illness, or living with someone who has a chronic illness, employment security, and educational sector. Conclusion: The study shows that it is important to attend to teachers’ health and quality of life, especially older teachers, those with a chronic illness, caregivers, those with job insecurity, and those who teach in preschool education.

## 1. Introduction

The COVID-19 pandemic is having an unprecedented impact on public health. Spain is one of the European countries that has so far been the most affected by COVID-19. Cases began to multiply exponentially and uncontrollably in early March 2020, prompting the Spanish government to declare a state of emergency on 14 March 2020, which included the closure of all schools in the country [1] along with the mandatory lockdown of all citizens [2]. As of 7 September 2020, 525,549 people had been diagnosed with COVID-19 in Spain, and 29,516 people have died of the illness in this country.

One of the most widespread measures taken to stop the expansion of COVID-19 was the closure of educational centers [3]. The United Nations Educational, Scientific, and Cultural Organization [4] estimated that, at the end of March 2020, 1,574,989,812 students (90% of those enrolled) in the world were affected by the closure of schools in response to COVID-19. Eight months later, with the beginning of the 2020–2021 school year, there were still 851,870,246 students in the world affected by school closures (48.7% of those enrolled).

In Spain, most schools reopened in September of the 2020–2021 school year. However, this reopening was somewhat controversial and full of uncertainties, as the Spanish government did not establish the minimum required conditions for the reopening of schools until 10 August. In addition, in some autonomous communities, those conditions were decided even later [5] (Rtve, 2020). For example, in the autonomous community of the Basque Country, these guidelines were agreed upon on 28 August, just ten days before the reopening of the schools on 7 September [6]. Similarly, the State Ministry of Universities disseminated its guidelines on 31 August.

Furthermore, the school community also complained that, aside from being late, some of the guidelines were ambiguous and left many decisions to be faced by the schools with hardly any new resources [7]. Therefore, it is clear that both the closure of schools and their reopening during the pandemic amid much uncertainty have affected millions of students and teachers, as they have had to adapt both their teaching format and their daily routines [8].

It should be borne in mind that, according to the World Health Organization [9], health is “a state of complete physical, mental and social wellbeing and not merely the absence of disease or infirmity” (p. 1), the study of how the current pandemic is affecting health should be analyzed both in terms of COVID-19 infections, as well as in terms of people’s quality of life at a holistic level. In the same vein, WHO also emphasizes that health is measured by the extent to which an individual or a group can change or cope with their environment [10].

Providing quality education has been defined as one of the objectives for the 2030 Agenda for Sustainable Development by the United Nations, and is therefore key for all countries [11]. In addition, education itself is also key to tackling the pandemic and reducing the social inequalities it may bring. Therefore, several researchers and experts have pointed out that the reopening of schools is essential to ensure the emotional, social, physical, and academic health of the population [12,13,14,15,16,17]. However, for quality education to be guaranteed, it is essential to ensure the health and wellbeing of the professionals who provide it, as UNESCO [4] has already identified that confusion and stress among teachers is one of the adverse consequences of school closures.

In other words, in addition to ensuring that schools are safe spaces on an epidemiological level, the quality of life of the professionals who work there must also be protected. According to the WHO, quality of life is defined in terms of how the individual perceives their cultural environment and the system of values in which they live with regard to certain objectives, criteria, and expectations. This is combined with physical health, psychological state, degree of independence, social relations, environmental factors, and personal beliefs.

Before the pandemic, the quality of life of teachers was identified as a key factor in ensuring quality education [18,19,20,21,22]. From these studies, it was concluded that environmental, contextual, and affective factors predominantly influence the overall wellbeing and quality of life of teachers [23]. The lack of psychological stability, social relations, and interaction with the environment can greatly impair the quality of life of these professionals. In addition, stressful work is responsible for the high risk of physical and psychological disorders, which can be detrimental to the workers’ quality of life [18,20].

In any case, the quality of life will vary among teachers and depend on several factors. To begin with, having secure employment is one of the clear variables that influences the perception of quality of life [24]. In addition, job insecurity has great psychological consequences, as shown in previous crises, and can significantly influence life quality.

Concerning gender, several studies indicate that, in general, women’s quality of life is often lower than men’s, both in the general population [25,26] and among teachers [27]. Likewise, in terms of age, studies show that the quality of life is poorer in older people in studies conducted in both non-COVID-19 times [28,29] and during the COVID-19 period [30]. However, in contrast to these findings, other studies indicate that younger people are showing more psychological suffering [31].

Moreover, as far as health variables are concerned, people who have illnesses such as heart disease, stroke, cancer, or lung disease experience a reduction in all levels of quality of life [32]. Indeed, a study conducted during the COVID-19 pandemic has shown that people with chronic illnesses are more likely to have a lower quality of life, as being more susceptible to the COVID-19 virus influences them in several aspects [30].

Framed within this context, by considering how teachers are coping with the return to school during this pandemic, we might be better positioned to put in place relevant support structures that may be needed to ensure quality education. Therefore, this research aims to study how teachers perceived their quality of life when coping with the reopening of schools in this new academic year during the COVID-19 crisis. Specifically, teachers’ perceived quality of life at the beginning of the 2020–2021 school year will be analyzed, considering the following four dimensions: physical health, psychological health, social relations, and environment. In addition, differences in quality of life will be analyzed according to gender, age, having a chronic illness or living with someone who has a chronic illness, employment security, and educational sector (infant education, primary education, high school, or university).

It is expected that teachers will have a lower perceived quality of life in social relations and the environment due to the direct influence that measures to stop COVID-19 may have had on them. It is also expected that women will have a lower perceived quality of life due to the care role that many have been fulfilling during the pandemic. Differences are also expected according to the age of the participants, with the older participants coping better due to their job security, among other reasons. It is also predicted that people with chronic diseases or those that live with people with chronic diseases will report a worse quality of life during a pandemic situation. Finally, job insecurity and the educational sector in which teachers work are also expected to affect the perceived quality of life, with the expectation that those working with the youngest children have to deal with more pressure and responsibility.

## 2. Materials and Methods

### 2.1. Participants

This study was carried out with a total sample of 1633 teachers from the Basque Autonomous Community (Northern Spain), who work in different educational stages. This sample was composed of 1293 women (M age = 42.6; SD = 9.96) and 330 men (M age = 42.02; SD = 10.40). The age range was 23–67 years. Of the sample, 18.9% were teaching in preschool education (*n* = 309), 32.55% in primary education (*n* = 530), 30.1% in secondary education (*n* = 491), 5.5% in bachelor studies (*n* = 89), 5.6% in vocational training (*n* = 91), and 7.5% in university studies (*n* = 123). A total of 47.3% (*n* = 772) of the participants had school-age children. In addition, 16.7% (*n* = 273) had a chronic disease, and 18.1% (*n* = 296) of the sample lived with someone with a chronic illness. Finally, 35.2% (*n* = 574) responded yes to the question of whether a family member had been infected with COVID-19.

### 2.2. Instruments

An ad hoc questionnaire was designed to collect data on personal, family, and social factors (gender, age, workplace, job stability, whether they had a chronic illness or lived with a person with a chronic illness).

Quality of life was measured using the Spanish version of WHOQOL-BREF (Espinoza et al., 2011), which is a short-form quality of life assessment. The WHOQOL-BREF focuses on the “perceived” quality of life of the participant. Therefore, the questionnaire evaluates a multidimensional concept that incorporates the individual’s perception of health and psychosocial status, along with other aspects of life. The scale consists of 26 items, which provides a profile composed of four dimensions: physical health, psychological health, social relations, and environment. Physical health includes the facets of activities of daily living, dependence on medicinal substances and medical aids, energy and fatigue, mobility, pain and discomfort, sleep and rest, and the ability to work (Items 2, 5, 6, 7, 15, 16, and 18). The following Likert-type responses were used: very dissatisfied, a little dissatisfied, normal, quite satisfied, and very satisfied. Psychological health includes facets of body image and appearance, negative feelings, positive feelings, self-esteem, spirituality/religion, personal beliefs and thinking, learning memory, and concentration (Items 4, 8, 17, 19, 24, and 26). Social relationships includes facets of personal relationships, social support, and sexual activity (Items 9, 10, and 11) both in the psychological and social dimension. The following Likert-type responses were used: nothing, a little, normal, quite, and extremely. Environment includes the facets of financial resources, freedom, physical safety, health and social care, accessibility and quality, home environment, opportunities to acquire new information and skills, participation and opportunities for leisure activities, physical environment (pollution/noise/traffic/climate) and means of transport (Items 12, 13, 14, 20, 21, 22, 23, and 25). The following Likert-type responses were used: very dissatisfied, a little dissatisfied, normal, quite satisfied, and very satisfied. Moreover, the following items were reversed: 15, 16, and 26. The higher the final score; the higher the quality of life. The minimum and maximum scores for each dimension in this study were the following: physical health: 4.80–16.80; psychological health: 5–16.67; social relationships: 4–16; and environment: 4–16.

Regarding the reliability of the scale, the Cronbach’s alpha coefficient was α = 0.85. For the physical health scale, α = 0.82; for the psychological health scale, α = 0.80; for the social relations dimension, α = 0.76; and for the environment dimension, α = 0.80.

### 2.3. Procedure

The sample was recruited through non-probabilistic sampling. An explanatory email with an online questionnaire to be filled in was sent to all schools in the region to reach all teachers. They were asked to disseminate this questionnaire among their teachers between 5–28 September 2020 (beginning of the school year). The questionnaire explained both the objectives of the study and the procedures to be followed during the study, as well as the right to voluntarily withdraw from the study if so desired. The Ethics Committee of the University of the Basque Country approved the study (UPV/EHU) (code M10/070/2020). For the data collection, all the canons established by the Organic Law 15/99 on the Protection of Personal Data were followed. In the questionnaires, participants were informed of the voluntary nature of their participation and the commitment required to start the test. Therefore, the procedure was carried out in accordance with the Declaration of Helsinki of the World Medical Association.

### 2.4. Data Analysis

The data were analyzed with the statistical program SPSS v.26 (IBM Corp., Armonk, NY, USA). First, the assumptions of normality and homoscedasticity of variances were checked to decide whether to use parametric or non-parametric tests. Specifically, the critical level of *p* < 0.05 was adopted for the Kolmogorov–Smirnov statistics, and the levels of asymmetry and kurtosis were analyzed. From these analyses, it was concluded that the data followed a normal distribution, so the authors decided to use parametric tests.

The dimensions of Quality of Life were categorized by the items indicated by the scale, and the items of the questionnaire with an inverse formulation were recorded. Subsequently, means were compared with the Student *t*-test when two groups were involved, and by univariate analysis, ANOVA, when analyzing the differences between more than two groups. The purpose of these analyses was to test for differences between the various quality of life dimensions and the socio-demographic variables analyzed for this study.

## 3. Results

### 3.1. General Characteristics of the Dimensions

The results indicate that the highest values of perceived quality of life were found for the dimension of psychological health (M = 11.48; SD = 2.06), followed by physical health (M = 11.20; SD = 2.02), social relations (M = 9.88; SD = 2.82), and finally the environmental dimensions (M = 9.77; SD = 1.86).

### 3.2. Significant Gender and Age Differences

Table 1 and Table 2 show gender differences in the social relations dimension, t (1621) = 2.64, *p* < *0*.008, Cohen’s d = 0.16, with women reporting the highest levels of quality of life in this area. Likewise, the environmental dimension, t (1621) = −2.42, *p* < *0*.016, Cohen’s d = 0.15, also differed significantly according to gender, but in this case men reported a greater environmental quality of life. Regarding age, there were statistically significant differences among all age ranges for all dimensions of the WHOQOL-BREF scale. For physical health, F (2, 1629) = 5.50, *p* = 0.04, η^2^ = 0.007, for psychological health, F (2, 1629) = 8.49, *p* = 0.01, η^2^ = 0.010, for social relations, F (2, 1629) = 12.18, *p* = 0.01, η^2^ = 0.015 and for environmental health, F (2, 1629) = 3.65, *p* = 0.26, η^2^ = 0.004. The differences (post hoc) among age groups are observable in Table 2.

### 3.3. Significant Differences According to Chronic Illness

Table 3 shows the descriptive statistics according to whether the participants had a chronic illness or were living with a chronically ill person. In Table 4, differences can be observed in terms of the various quality of life dimensions. In this case, the psychological health dimension, t (1631) = −3.84, *p* < 0.001, Cohen’s d = 0.25 differs significantly according to whether or not the participants are chronically ill, as those without a chronic illness reported a greater psychological quality of life. Moreover, differences in the environmental quality of life dimension, t (1631) = −3.80, *p* < 0.001, Cohen’s d = 0.24, show that those without a chronic illness have a higher perceived quality of life. Differences in both dimensions are with a small effect size. In the case of living with a chronically ill person, the differences are statistically significant for all dimensions. Participants who do not live with a chronically ill person show a higher perceived quality of life (See Table 4).

### 3.4. Significant Differences According to Job Stability and Workplace

Concerning the job stability variable, statistically significant differences were again found in the psychological, F (2, 1629) = 2.65, *p* = 0.047, η^2^ = 0.005 and environmental dimensions, F (2, 1629) = 3.08, *p* = 0.026, η^2^ = 0.006, both with small effect sizes. The differences for this dimension of psychological quality of life were found among people working for less than three months and people with indefinite contracts, with people with indefinite contracts showing a higher perceived psychological quality of life. The same occurs in the dimension of environmental quality of life, showing statistically significant differences between teachers with contracts of less than three months compared to those with indefinite contracts, and those with contracts of between three months and one year compared to those with indefinite contracts (See Table 5 and Table 6).

Finally, concerning the educational sector where the participants work, there are statistically significant differences in the environmental dimension, F (2, 1627) = 3.70, *p* = 002, η^2^ = 0.011, showing differences in scores between those working in preschool education and those in primary education (higher scores for primary education teachers), between the preschool education and high school teachers (higher scores for high school teachers) and between preschool education and university teachers (higher scores for university teachers). However, in these cases, the effect sizes were small (See Table 5 and Table 6).

## 4. Discussion

The main objective of this research was to study the quality of life of teachers at the beginning of the 2020–2021 school year during a pandemic. The results have revealed that teachers have shown the highest levels of perceived quality of life in the psychological health dimension, followed by the physical health dimension, the social relationships dimension, and the environmental dimension.

First, the environmental dimension includes factors such as the perception of freedom, physical security, health, and social care, the domestic environment, opportunities to acquire new information and skills or participation, and opportunities for leisure activities [33,34]. All these factors have been affected by restrictive measures and social distancing. Moreover, health and social care could also be affected by the collapse of the health system during the pandemic [35]. However, studies conducted before the pandemic also showed lower scores in the environmental domain [18,20,36]. Therefore, although the pandemic may have influenced this domain, teachers already perceived low environmental quality of life [36].

Second, the social relations dimension includes facets of personal relationships, social support, and sexual activity [33,34]. Among these, interpersonal relationships have been restricted by the measures implemented to cope with COVID-19. When this study was carried out in Spain, people could congregate in groups of no more than 10 people. Further, loneliness and lack of social support have been characteristic emotions experienced during this pandemic, particularly during lockdown [37]. In studies conducted before the pandemic, teachers scored highest in the area of social relationships, and therefore it is possible that the pandemic has had an impact on this dimension [38,39].

Third, the physical health dimension includes factors such as activities of daily living, energy and fatigue, sleep and rest, and ability to work [33,34]. It should be noted that the WHO has already highlighted pandemic fatigue as one of the consequences of COVID-19 [9,40], and several studies suggest that the pandemic has led to an increase in sleep problems [41,42]. Finally, regarding the ability to work, we must remember that teachers had to adapt first to an online teaching modality in record time [43,44,45], and now face a new academic year full of uncertainty regarding their teaching activity [7]. However, the present study suggests that the physical dimension of the teachers is one of the least threatened, compared with previous scores [27,39], so it would not be a major concern for this population.

Fourth, this study indicates that it is in the dimension of psychological health where the greatest quality of life is perceived. Although, in this scale, emotional health is measured from a very wide prism, we must point out that in other studies carried out with the same sample, it was found that these teachers showed worrying levels of stress, anxiety, and depression. For example, 50.6% of teachers indicated that they suffer from stress, 49.5% from anxiety, and 32.2% from depression [46].

The results have also revealed significant differences in perceived quality of life depending on personal and professional variables. Although most studies in the literature have pointed out that men usually have a higher perception of quality of life than women [26,27,47], this study has shown that men only have a higher perception of quality of life in the environmental dimension. Women even show a greater perceived quality of life in the social relationships dimension. Other studies conducted before the pandemic with teachers also reported a poorer quality of life among women [18].

Therefore, the fact that women showed a higher perceived quality of life in the dimension of social relationships could be as feelings of isolation and loneliness during this pandemic have particularly affected women [31,48], so the return to work may have alleviated some of those symptoms. That is, for them, it has been a greater relief to be able to meet their peers again, as these interpersonal relationships relieve stress and provide support and motivation to cope with the pandemic situation. In addition to analyzing these results, the uniqueness of the sample must also be taken into account, given the higher predominance of females in the teaching profession.

Regarding age, similar to previous studies, a significantly worse perception of quality of life has been found in older people [30]. Our findings are in accord with a study conducted in China, at least in the psychological quality of life dimension. In this study, the older teachers showed the greatest levels of psychological distress [49]. Our study confirms that older teachers perceive a worse quality of life beyond the psychological level in the dimension of physical health and social relations. However, surprisingly, they have the highest perceived quality of life in the environmental dimension. One of the reasons why older teachers may have a worse perceived quality of life could be as, in this new situation, the adaptability to new technologies is essential, and this group of teachers are usually less interested and have lower-level skills in Information and Communication technologies [50]. In studies conducted before the pandemic, older teachers had shown better [51] or similar [27] quality of life to other teachers. Therefore, it can also be concluded that the pandemic may have affected the perceived quality of life of older teachers.

Concerning differences according to chronic illness, teachers who live with a chronically ill person have worse self-perception of quality of life than those who do not. This was found for all quality of life dimensions. These results were expected, as people with chronic diseases have been the most vulnerable in the current pandemic [52], so teachers may be afraid of infecting their families. What is remarkable is that the same results were not found among people with chronic diseases. In other words, people with chronic illness have only perceived significantly lower quality of life in psychological health and environment dimensions. Therefore, caregivers must be considered, particularly in times of a pandemic, as their quality of life can be substantially affected. They would therefore need special support, as noted in previous studies [53,54]. The results also show that the highest perceived quality of life in the dimensions of psychological health and environment is shown by people with indefinite contracts. Several investigations have analyzed the impact of job insecurity on teachers, and have pointed out that this factor can have important psychological consequences. Moreover, it has also been demonstrated that persistent and long-term unemployment and precarious employment cause a significant deterioration in the perception of environmental quality of life [24]. Likewise, the world economic crisis derived from COVID-19 is putting many jobs at risk, and lack of secure employment is becoming an aggravated problem in Spain [55].

Concerning the educational sector where teachers work, it should be considered that face-to-face teaching was resumed in the 2020–21 academic year in Spain. However, each of the academic sectors established a different teaching scenario. On the one hand, compulsory basic education and secondary education adopted very restrictive safety measures, but opted for fully face-to-face teaching. On the other hand, university education had instructions for semi-face-to-face, bimodal, or even online training in some cases. Therefore, these particularities are factors that could make teachers’ perceptions vary, depending on the academic level at which they teach. Specifically, it was found that preschool teachers have shown lower levels of perceived quality of life in the environmental dimension. Given that this quality of life dimension includes factors such as physical safety and health safety, it may be that teachers who work with the youngest children and in a fully face-to-face mode do not perceive their work to be safe, especially when compared to university teachers who have much less contact, as in many cases, classes were relaunched in an online or bimodal mode. This may also be as children under five years are not obliged to use masks in Spain, and it may also be more difficult for these children to understand and follow hygiene and social distancing measures [56]. However, it should be remembered that infections among younger children are fewer [57], although it is logical that teachers may feel this lack of safety due to all the uncertainty and unknowns that this pandemic has brought.

One of the strengths of the present study is that, to the best of our knowledge, it is the first study to examine the quality of life of teachers in the reopening of schools after the lockdown period and during the COVID-19 pandemic. Moreover, this research also has some relevant practical implications. First, as long as restrictive measures remain active to stop the pandemic, it will be important to redefine educational centers (schools, high-schools, and universities) as spaces where physical safety, health, and social care, accessibility, and opportunities to acquire new information and skills and participate in leisure activities are reinforced for teachers. In addition, although it is necessary to continue to maintain social distancing, it would also be relevant to try to promote personal relations and especially social support by putting in place, for example, professional online support networks.

Finally, beyond preventing infections, teachers’ health and wellbeing will also be linked to the physical and psychological dimensions examined in this study. In other words, overall wellbeing must be promoted from a holistic perspective among teachers, reinforcing actions for older teachers, the chronically ill, and those who are caregivers or work in preschool education. In the same vein, to ensure the quality of life of teachers, it would also be important to guarantee their job stability.

However, this study also has several limitations. First, a non-probabilistic sample was recruited, which may have a certain selection bias as participation was voluntary, and it is likely that only those who were particularly emotionally impacted would have participated. Therefore, future studies should include a more balanced probabilistic sample concerning gender and recruit participants from other autonomous communities. Second, this is also a cross-sectional study, and further follow-up will help us to draw more valid conclusions. Third, relatively few studies have been conducted on the pre-pandemic quality of life of Spanish teachers. This has limited the possibility of comparing previous results with current outcomes. Finally, no studies have been conducted with the general population in Spain using the WHOQOL-Brief instrument, limiting the comparison of our findings with those of other studies.

## 5. Conclusions

From the results presented in this article, it can be concluded that teachers’ quality of life in a pandemic differs according to the dimension analyzed, along with both personal and professional characteristics. In the current COVID-19 pandemic context, ensuring health and wellbeing is key at a global level. However, to achieve the wellbeing of the population, a strong and healthy education system is necessary. In this equation, the quality of life of teachers is crucial. That is, the health status and wellbeing of teachers are key to their ability to support the wellbeing of their students, and this assumption makes it essential to recognize the importance of the quality of life of teachers during this pandemic. Therefore, it is important to take measures and provide resources for teachers to increase their wellbeing in the dimensions of physical health, psychological health, social relations, and environment, paying special attention to those teachers who are most vulnerable. Such measures could then serve to protect the health and wellbeing of both teachers and students, which, in turn, would help to develop and maintain quality education. In this regard, research that examines the specific needs of teachers in this unprecedented situation is fundamental for developing realistic strategies.

## Figures and Tables

**Table 1 ijerph-18-07791-t001:** Means and standard deviations in quality-of-life scores according to gender and age.

Dimension—WHOQOL-BREF				
Gender		*n*	Mean	Standard Derivation
Physical health	Female	1293	11.17	2.02
	Male	330	11.35	2.07
Psychological health	Female	1293	11.46	2.03
	Male	330	11.56	2.17
Social relationships	Female	1293	9.98	2.77
	Male	330	9.52	2.97
Environment	Female	1293	9.71	1.86
	Male	330	9.99	1.98
Age				
Physical health	23–35	451	11.47	2.16
	36–45	578	11.11	1.92
	>46	603	11.09	1.99
Psychological health	23–35	451	11.79	2.09
	36–45	578	11.45	2.02
	>46	603	11.27	2.04
Social relationships	23–35	451	10.43	3.00
	36–45	578	9.80	2.78
	>46	603	9.58	2.66
Environment	23–35	451	9.86	1.93
	36–45	578	9.60	1.90
	>46	603	9.87	1.83

**Table 2 ijerph-18-07791-t002:** Student’s *t*-test results according to gender and age.

Dimensions WHOQOL−BREF					
Gender	t	df	*p*	d	
Physical health	−1.44	1621	0.151		
Psychological health	−0.798	1621	0.425		
Social relationships	2.64	1621	0.008 *	0.16	
Environment	−2.42	1621	0.016 *	−0.15	
Age	F	df	*p*	η^2^	Post hoc
Physical health	5.50	2	0.004 **	0.007	1−2;1−3
Psychological health	8.49	2	0.001 ***	0.010	1−2;1−3
Social relationships	12.18	2	0.001 ***	0.015	1−2;1−3
Environment	3.65	2	0.026 *	0.004	1−3;2−3

Note: *** *p* < *0*.001; ** *p* < 0.01; * *p* < 0.05.

**Table 3 ijerph-18-07791-t003:** Means and standard deviations in quality-of-life scores according to whether the participants have a chronic illness or are living with a chronically ill person.

Dimension—WHOQOL-BREF				
Chronic Illness		*n*	M	SD
Physical health	Yes	273	11.00	1.93
	No	1360	11.24	2.04
Psychological health	Yes	273	11.04	2.15
	No	1360	11.56	2.02
Social relationships	Yes	273	9.64	2.86
	No	1360	9.93	2.81
Environment	Yes	273	9.37	1.86
	No	1360	9.85	1.88
Living with a chronically ill person		*n*	M	SD
Physical health	Yes	296	10.83	2.06
	No	1337	11.29	2.00
Psychological health	Yes	296	10.94	2.19
	No	1337	11.59	2.01
Social relationships	Yes	296	9.29	2.65
	No	1337	10.02	2.84
Environment	Yes	296	9.09	1.82
	No	1337	9.92	1.87

**Table 4 ijerph-18-07791-t004:** Student’s t test results according to whether participants have a chronic illness or are living with a chronically ill person.

Dimensions WHOQOL−BREF				
Chronic Illness	t	df	*p*	d
Physical health	−10.82	1631	0.068	0.12
Psychological health	−30.84	1631	0.001 ***	0.25
Social relationships	−10.61	1626	0.106	0.10
Environment	−30.80	1631	0.001 ***	0.24
Living with a chronically ill person	t	df	*p*	d
Physical health	−30.50	1631	0.001 ***	0.23
Psychological health	−40.91	1631	0.001 ***	0.32
Social relationships	−40.06	1631	0.001 ***	0.26
Environment	−70.02	1631	0.001 ***	0.45

Note: *** *p* < 0.001.

**Table 5 ijerph-18-07791-t005:** Means and standard deviations in quality-of-life scores according to length of contract in months (job stability) and workplace.

Dimension—WHOQOL-BREF				
Months of Contract		*n*	M	SD
Physical health	Less than three months	54	11.10	2.12
	Between 3 months and one year	102	1.086	2.31
	One year	503	11.15	2.01
	Indefinite contract	974	11.28	1.98
Psychological health	Less than three months	54	10.98	2.20
	Between 3 months and one year	102	11.53	2.00
	One year	503	11.33	2.08
	Indefinite contract	974	11.57	2.04
Social relationships	Less than three months	54	10.17	2.98
	Between 3 months and one year	102	9.91	3.12
	One year	503	9.79	2.95
	Indefinite contract	974	9.92	2.71
Environment	Less than three months	54	9.29	1.60
	Between 3 months and one year	102	9.46	1.91
	One year	503	9.71	1.97
	Indefinite contract	974	9.87	1.85
	Educational sector	*n*	M	SD
Physical health	Preschool Education	309	11.14	1.95
	Primary Education	530	11.28	2.04
	Obligatory Secondary Education	491	11.13	2.04
	Sixth Form College	89	11.65	1.79
	University studies	123	10.97	2.09
	Vocational training	91	11.30	2.14
Psychological health	Preschool Education	309	11.34	2.00
	Primary Education	530	11.60	2.03
	Compulsory Secondary Education	491	11.37	2.05
	Sixth Form College	89	11.72	2.19
	University studies	123	11.61	2.12
	Vocational training	91	11.42	2.30
Social relationships	Preschool Education	309	9.78	2.79
	Primary Education	530	10.09	2.78
	Compulsory Secondary Education	491	9.85	2.82
	Sixth Form College	89	9.81	2.97
	University studies	123	9.41	2.69
	Vocational training	91	9.96	3.10
Environment	Preschool Education	309	9.45	1.91
	Primary Education	530	9.76	1.81
	Compulsory Secondary Education	491	9.78	1.86
	Sixth Form College	89	10.21	2.17
	University studies	123	10.13	1.82
	Vocational training	91	9.92	1.98

**Table 6 ijerph-18-07791-t006:** Student’s *t*-test results according to length of contract in months (job stability) and workplace.

Dimensions WHOQOL-BREF					
Months of Contract	F	df	*p*	η^2^	Post hoc
Physical health	1.54	1629	0.202	0.003	
Psychological health	2.65	1629	0.047	0.005	1–4
Social relationships	0.415	1624	0.743	0.001	
Environment	3.08	1629	0.026	0.006	1–4;2–4;
Educational Sector	F	df	*p*	η^2^	Post hoc
Physical health	1.59	1627	0.160	0.004	
Psychological health	1.28	1627	0.268	0.004	
Social relationships	1.41	1622	0.218	0.003	
Environment	3.70	1627	0.002	0.011	1–2;1–4

## Data Availability

The data that support the findings of this study are available on request from the corresponding author, N.O.-E. The data are not publicly available due to containing information that could compromise the privacy of research participants.

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
