# Peer review of "Reopening of Schools in the COVID-19 Pandemic: The Quality of Life of Teachers While Coping with This New Challenge in the North of Spain"

_ijerph, 2021, doi:10.3390/ijerph18157791_

Round 1

Reviewer 1 Report

Some changes and improvements are needed in order to satisfy Journal quality standards:

Ln 58, and Ln. 81.  delete the words " In fact" and start the sentence with the rest of sentence.  Because, words "in fact" apear too frequently in this part of text.

Ln. 115 "It is expected that teachers will have a lower perception of quality of life in the…"  omit "perception of"

Ln. 142 "The scale used for the analysis of quality of life was…"  change to:  "The quality of life was measured by Spanish version…"

Ln. 147 "which includes a profile…"  change to: "which provide a profile…"

In section 2.2. Instruments:

Provide the information on theoretical range, minimal and maximal scores for WHOQOLBref dimensions. This was needed for readers to easier understand how high or low is mean score of each dimension.

In table 2:  do not show "d" in header row when there is no values under that header.

In table 2: Post-hoc column – is it needed to repeat same difference just opposite direction? 1-2  and 2-1 ? Redundant, advise to correct in order indicating difference between groups just once, for example, for physical health post hoc 1-2; 1-3.  Check all the tables with post hoc.

In Discussion:

Ln. 319  "…person have worse quality of life outcomes than those…"  it is not Outcomes that was measured, but own QOL perception, so omit the word outcomes and rewrite this part of sentence.

 Ln 337 and Ln 358 starts with "Finally..", change in a way that just final statement starts with finally, not both.

Improvement of Discussion needed: indicate that previous QOL of those participants was not known, so it can not be said that COVID decreased their QOL.  However, the topic - QOL of teachers is relevant for wider audience but discussion needs improvement, clearly explaining that these results cannot be attributed to pandemic or lockdown itself. Try to rewrite text to maintain the focus on teachers QOL levels but in interpretation will be beneficial to compare these results with previous population studies which uses WHOQOL-Bref instrument. What was QOL of similar population groups before the pandemic? For example, in samples of open adult population, or people in education sector in Country or EU.

Also, in section mentioning limitation, point out that this is cross sectional research and further follow up will be beneficial for more valuable conclusions.

Author Response

Dear reviewer,

We would like to thank you for considering our manuscript for publication in International Journal of Environmental Research and Public Health. We have found your comments and insights to be most constructive and helpful and we have tried to address the issues you proposed.

We attach to this letter the word document with change “change editor” tool, where all the changes that we have made are. Moreover, in the following text we will detail, reviewer by reviewer, the changes that have been made.

Thank you very much for your comments and the detail of your suggestions. We have responded to all of them and we have also sent the article back to a professional native English proofreader to improve its English writing.

  • Ln 58, and Ln. 81.  delete the words " In fact" and start the sentence with the rest of sentence.  Because, words "in fact" apear too frequently in this part of text.

We have deleted them

  • 115 "It is expected that teachers will have a lower perception ofquality of life in the…"  omit "perception of"

We have deleted it

  • 142 "The scale used for the analysis of quality of life was…"  change to:  "The quality of life was measured by Spanish version…"

We have changed it

  • 147 "which includes a profile…"  change to: "which provide a profile…"

We have changed it

In section 2.2. Instruments:

  • Provide the information on theoretical range, minimal and maximal scores for WHOQOLBref dimensions. This was needed for readers to easier understand how high or low is mean score of each dimension.

We have provided the minimum and maximum scores of each dimension

  • In table 2:  do not show "d" in header row when there is no values under that header.

We have added the values  of “d”

  • In table : Post-hoc column – is it needed to repeat same difference just opposite direction? 1-2  and 2-1 ? Redundant, advise to correct in order indicating difference between groups just once, for example, for physical health post hoc 1-2; 1-3.  Check all the tables with post hoc.

We have eliminated the redundancies

In Discussion:

  • 319  "…person have worse quality of life outcomes than those…"  it is not Outcomes that was measured, but own QOL perception, so omit the word outcomes and rewrite this part of sentence.

We have rewritten it

  • Ln 337 and Ln 358 starts with "Finally..", change in a way that just final statement starts with finally, not both.

We have rewritten it

  • Improvement of Discussion needed: indicate that previous QOL of those participants was not known, so it can not be said that COVID decreased their QOL.  However, the topic - QOL of teachers is relevant for wider audience but discussion needs improvement, clearly explaining that these results cannot be attributed to pandemic or lockdown itself. Try to rewrite text to maintain the focus on teachers QOL levels but in interpretation will be beneficial to compare these results with previous population studies which uses WHOQOL-Bref instrument. What was QOL of similar population groups before the pandemic? For example, in samples of open adult population, or people in education sector in Country or EU.

We have rewritten the entire discussion section incorporating new references to compare our findings with previous research on teachers' quality of life point by point.

  • Also, in section mentioning limitation, point out that this is cross sectional research and further follow up will be beneficial for more valuable conclusions.

We have included it in the limit section

Reviewer 2 Report

The paper was clearly written and easy to follow. It sets out and highlights some of the pressures and challenges that teachers face due to the specific nature and demands of their profession.

LINES 370-381 In the conclusion: I agree that examining the specific needs of teachers is important. However, I felt the study underplayed the connection between teachers' ability to support the mental and physical health and wellbeing of their students and their own states of health and wellbeing. Perhaps I could describe it like this: attend to yourself so that you are in a position to help others. I think a connection such as this could be made more clearly. The first couple of lines (371-373) of the conclusion could be re-written to achieve this. 

The WHO definition of health and wellbeing is used in the introduction - I think keep this consistent throughout the paper, i.e. in the conclusion and elsewhere (for example, in line 71) talk about "health and wellbeing" or "overall wellbeing" rather than just "health." You explain this well in lines 74-79. Be consistent throughout in the use of terms and make clear the links between "quality of life" and the dimensions of "health" referred to, and "wellbeing".

Author Response

Dear reviewer,

We would like to thank you for considering our manuscript for publication in International Journal of Environmental Research and Public Health. We have found your comments and insights to be most constructive and helpful and we have tried to address the issues you proposed.

We attach to this letter the word document with change “change editor” tool, where all the changes that we have made are. Moreover, in the following text we will detail, reviewer by reviewer, the changes that have been made.

Thank you very much for your comments and suggestions. Following your suggestions we have rewritten both the conclusion section and revised the conception of health throughout the article. In addition, we have also re-sent the article to a professional native English proofreader to improve the English wording.

  • LINES 370-381 In the conclusion: I agree that examining the specific needs of teachers is important. However, I felt the study underplayed the connection between teachers' ability to support the mental and physical health and wellbeing of their students and their own states of health and wellbeing. Perhaps I could describe it like this: attend to yourself so that you are in a position to help others. I think a connection such as this could be made more clearly. The first couple of lines (371-373) of the conclusion could be re-written to achieve this. 

We have rewritten the part of the conclusions to emphasise the ideas proposed by the reviewer.

  • The WHO definition of health and wellbeing is used in the introduction - I think keep this consistent throughout the paper, i.e. in the conclusion and elsewhere (for example, in line 71) talk about "health and wellbeing" or "overall wellbeing" rather than just "health." You explain this well in lines 74-79. Be consistent throughout in the use of terms and make clear the links between "quality of life" and the dimensions of "health" referred to, and "wellbeing".

We fully agree that it is necessary to include this holistic conception of health and have therefore revised the entire article and incorporated the ideas proposed by the reviewer.

Reviewer 3 Report

The work titled Reopening of schools in the COVID-19 pandemic: The quality 2 of life of teachers while coping with this new challenge in the 3 north of Spain, addresses a very current topic of great social interest.

I would like to congratulate the authors for the work that I consider to be well planned and carried out, but I would like the authors to reflect the following considerations in the document.

In the Spanish nation, face-to-face teaching was resumed in academic year 20-21, but each of the academic degrees established a different teaching scenario. On the one hand the compulsory basic training took some security measures, and on the other hand the secondary training, despite the fact that both opted for total pre-specialization. However, the university training, had instructions to carry out a blended training. For this reason, I consider that these particularities are factors that make the perception of each of the teachers vary, according to the academic degree.

I believe that the authors did not delve into this matter, I invite them to do so and reflect it in the work, especially in the discussion section.

On the other hand, I recommend that you reformulate the conclusions in such a way that they answer the objectives clearly and directly.

Author Response

Dear reviewer,

We would like to thank you for considering our manuscript for publication in International Journal of Environmental Research and Public Health. We have found your comments and insights to be most constructive and helpful and we have tried to address the issues you proposed.

We attach to this letter the word document with change “change editor” tool, where all the changes that we have made are. Moreover, in the following text we will detail, reviewer by reviewer, the changes that have been made.

Thank you very much for your comments and suggestions. Following your suggestions we have rewritten both the discussion and the conclusions sections.

  • In the Spanish nation, face-to-face teaching was resumed in academic year 20-21, but each of the academic degrees established a different teaching scenario. On the one hand the compulsory basic training took some security measures, and on the other hand the secondary training, despite the fact that both opted for total pre-specialization. However, the university training, had instructions to carry out a blended training. For this reason, I consider that these particularities are factors that make the perception of each of the teachers vary, according to the academic degree. I believe that the authors did not delve into this matter, I invite them to do so and reflect it in the work, especially in the discussion section.

 We have incorporated this very interesting reflection in the discussion section.

  • On the other hand, I recommend that you reformulate the conclusions in such a way that they answer the objectives clearly and directly.

The conclusions section has been reformulated on the basis of the research objectives but also taking into account the suggestions of the other reviewers.

Round 2

Reviewer 1 Report

Considering changes and improvements suggested by all reviewers, and which authors made/included in text in second version, I recommend publishing of this paper.